# Serial Cultivation of an MSC-Like Cell Line with Enzyme-Free Passaging Using a Microporous Titanium Scaffold

**DOI:** 10.3390/ma16031165

**Published:** 2023-01-30

**Authors:** Yukihiko Sakisaka, Hiroshi Ishihata, Kentaro Maruyama, Eiji Nemoto, Shigeki Chiba, Masaru Nagamine, Hiroshi Hasegawa, Takeshi Hatsuzawa, Satoru Yamada

**Affiliations:** 1Department of Periodontology and Endodontology, Tohoku University Graduate School of Dentistry, 4-1 Seiryou-machi, Aoba-ku, Sendai 980-8575, Japan; 2Nagamine Manufacturing Co., Ltd., 1725-26, Kishinoue, Manno-cho, Nakatado-gun, Kagawa 766-0026, Japan; 3Department of Oral Surgery and Dentistry, Fukushima Medical University, 1, Hikariga-oka, Fukushima 960-1295, Japan; 4Laboratory for Future Interdisciplinary Research of Science and Technology, Institute of Innovative Research, Tokyo Institute of Technology, 4259 Nagatsuta-cho, Midori-ku, Yokohama, Kanagawa 226-8503, Japan

**Keywords:** microporous titanium, surface morphology, cell cultivation, scaffolds, passage, C3H10T1/2

## Abstract

In vitro studies on adherent cells require a process of passage to dissociate the cells from the culture substrate using enzymes or other chemical agents to maintain cellular activity. However, these proteolytic enzymes have a negative influence on the viability and phenotype of cells. The mesenchymal stem cell (MSC)-like cell line, C3H10T1/2, adhered, migrated, and proliferated to the same extent on newly designed microporous titanium (Ti) membrane and conventional culture dish, and spontaneous transfer to another substrate without enzymatic or chemical dissociation was achieved. The present study pierced a 10 μm-thick pure Ti sheet with 25 μm square holes at 75 μm intervals to create a dense porous structure with biomimetic topography. The pathway of machined holes allowed the cells to access both sides of the membrane frequently. In a culture with Ti membranes stacked above- and below-seeded cells, cell migration between the neighboring membranes was confirmed using the through-holes of the membrane and contact between the membranes as migration routes. Furthermore, the cells on each membrane migrated onto the conventional culture vessel. Therefore, a cell culture system with enzyme-free passaging was developed.

## 1. Introduction

Cultivating cells on a smooth substrate is a fundamental principle in the culture of adherent cells. In a conventional method, the cells migrate to and proliferate in vacant areas pretreated for cell adhesion in the substrate. When the culture stage approaches confluence, the cells cover the entire surface of the culture dish, and the lack of space for growth suppresses proliferation. Enzymatic passages with dissociation protocols, in which trypsin with or without EDTA is typically used [1,2], are the most common methods employed to isolate the cells from confluent conditions and expand cultures. However, the dissociation using proteolytic enzymes may impair the viability and phenotype of the cells due to technical sensitivity [3,4]. Overtreating the cells with enzymatic digestions could degenerate proliferation capacity and influence their ability to differentiate into mature cells [5]. Notably, human embryonic stem cells (hESCs) are influenced by genetic and epigenetic stability [6] and are susceptible to apoptosis induced by enzymatic detachment and dissociation [7]. Furthermore, the enzymatic passaging decreased glucose oxidation and fatty acid synthesis of hESCs [8]. Other passaging methods to detach adherent cells from the substrate employ chemical agents [9,10] or physical harvesting by shaking or scraping the culture vessel [11]. However, they risk influencing cell metabolism and decreasing cell viability [12]. The processes to isolate and transfer cells have yet to be developed for living biotissues. A cell detachment procedure using trypsin reduces the antigenicity of the surface markers of the stem cells [13]. Therefore, maintaining cell proliferation without dissociation is essential for evaluating biological characteristics under normal tissue conditions.

Pure titanium (Ti), which exhibits superior biocompatibility, is commonly used as a scaffold with mesenchymal stem cells (MSC) for osteogenic regeneration therapy in the medical field [14,15]. We previously investigated the proliferation and differentiation of the cells cultivated on a membrane-styled Ti scaffold with unified micron-sized pore arrays [16,17]. A 10 μm-thick microporous Ti membrane has a plane substrate for culturing and pathways for the migration of cells to the opposite side, suggesting that the cells migrate from the Ti membrane to the stacked vacant substrates. The cell proliferation in multi-layered Ti membranes with spatial diversity may apply to non-enzymatic cultures without dissociation from the substrate.

This study compared the structural features of Ti membranes between the single-layered 2-dimensional (2D) culture and multi-layered 3D-like scaffold and observed the proliferation of the C3H10T1/2 cells, a mouse MSC-like progenitor cell line [18], under both conditions.

## 2. Materials and Methods

### 2.1. Preparation of a Microarrayed Pin Needle Punch (MPNP) and Micropiercing

An MPNP was prepared from the plane surface of a cemented carbide metal block (Super ultra-fine grain carbide “AF1”: Sumitomo Electric Hardmetal, Itami, Japan). The surface was carved using a diamond dicing wheel with a punching die on which square pins with a side length of 25 microns were arranged in a 2D lattice array at 75 μm intervals [16]. The surface of the punch with a microneedle array included approximately 17,700 pins within a 10 mm square. The fabrication process with MPNP was performed for piercing through the 10 μm-thick Ti sheet (Figure 1).

### 2.2. Cell Culture

C3H10T1/2 cells, a murine immortalized MSC-like cell line, was purchased from the American Type Culture Collection (Manassas, VA, USA) and maintained in Dulbecco’s modified Eagle’s medium (Gibco^TM^/Thermo Fisher Scientific, Waltham, MA, USA) containing 10% fetal bovine serum (Biowest, Nuaillé, France). The cells were seeded at a density of 50,000 cells/cm^2^ in 12-well culture plates with Ti membranes and incubated. The medium was changed every 3 days. To evaluate the cell growth on both sides of the Ti membrane, the cells were seeded on the head or tail side in the well and incubated for 5 days. The cells grew and almost filled the micropores on both sides of the substrate after the 5-day culture. To assess the migration of cells through the micropores and to the neighboring Ti membranes, the cells were seeded on the head side. The day after the first 24 h of seeding, the Ti membrane with cells was placed between 4 new membranes, 2 upper and 2 lower, without pressing and incubated for 10 days (first multi-layered culture) (Figure 2a–d). The substrate with seeded cells was defined as “±0”. The layers above “±0” were defined as positive numbers (+1, +2), whereas the lower layers were negative numbers (−1, −2). After the first multi-layered culture, layers +2 and −2 were separately placed between 4 new layers and incubated for 10 days (second multi-layered culture) (Figure 2e–g). The substrates defined as “+2” and “−2” in the first multi-layered culture were used as “+2 ± 0” and “−2 ± 0” in the second multi-layered culture. The substrates sandwiching “+2 ± 0” or “−2 ± 0” were assigned with positive or negative numbers, respectively. To evaluate the migration of the cells on the Ti membrane to a conventional culture dish, the highest and lowest layers following the second multi-layered culture were transferred into a new 12-well culture plate and incubated (Figure 2h). After a 3-day culture, Ti membranes were removed, and the cells migrating to the culture plate were incubated for 6 days.

### 2.3. Scanning Electron Microscopy (SEM) Observations of the Cell Morphology

The cells cultured on Ti membrane over a 5-day period were fixed in 2% (*w*/*v*) glutaraldehyde in phosphate-buffered saline (PBS) (Nacalai Tesque, Kyoto, Japan) and incubated at 4 °C for 1 h. After washing with PBS, the specimens were dehydrated by washing in increasing concentrations of ethanol. Following immersion in t-butyl alcohol (FUJIFILM Wako Pure Chemical, Osaka, Japan) for 30 min, the specimens were lyophilized with a critical point dryer (VFD21-S) (VACUUM DEVICE, Ibaraki, Japan). The dried samples were mounted on aluminum stages using double-sided tape and coated with platinum using an ion sputter coater (JFC-1600) (JEOL, Tokyo, Japan). All specimens were subjected to SEM observations (JSM-6390LA) (JEOL).

### 2.4. Immunofluorescence Assay

The cells cultured on Ti membrane were fixed with 2% (*w*/*v*) paraformaldehyde (Nacalai Tesque) for 15 min, incubated with PBS containing 0.25% Triton X-100 (Sigma-Aldrich^®^/Merck KGaA, Darmstadt, Germany) (PBS-T) for 10 min to permeabilize cells, and incubated in PBS-T containing 1% (*w*/*v*) bovine serum albumin (Sigma-Aldrich^®^/Merck KGaA) for 30 min to block non-specific binding. The cells were incubated with a 1:20 dilution of Alexa Fluor^®^ 488-conjugated phalloidin (Cell Signaling Technology, Danvers, MA, USA) for 1 h. After incubating with DAPI (Invitrogen^TM^/Thermo Fisher Scientific) for 1 min to identify the nuclei, the staining of F-actin was evaluated by immunofluorescence microscopy (Leica M165FC) (Leica Microsystems, Nussloch, Germany). All procedures were performed at room temperature. The images were converted to binary using GIMP 2.8.10. The stained areas of nuclei were quantified.

### 2.5. Statistical Analyses

All experiments in the present study were performed thrice to test the result’s reproducibility, and the representative results are shown. All values are expressed as the mean ± standard error of the mean. All results were compared using a one-way analysis of variance followed by Tukey’s test. The differences with *p*-values < 0.05 were considered significant.

## 3. Results

### 3.1. Fabrication of a Microperforated Ti Membrane

The photographs and SEM images of a microperforated Ti membrane after MPNP processing are shown in Figure 3. High-density micropores enabled the visualization of the letter behind the Ti membrane (Figure 3b). The shapes of the perforated apertures were uniform at 25 μm × 25 μm and arrayed at 75 μm intervals. We defined the head side as the side contacted by the MPNP and the tail side as the opposite side. The surface of the head side contacted by the MPNP remained smooth, and square holes with the edges of apertures were straight and sharp (Figure 3c,d). On the other hand, the shape of each hole on the tail side was generally uniform at 25 μm × 25 μm. In contrast, the penetrated part of the substrate was accompanied by irregularly shaped edges with burrs mimicking a “volcanic” shape (Figure 3e,f). The 0.5 mm edge of the Ti membrane was bent toward the head side to be easily held by tweezers.

### 3.2. Morphology of C3H10T1/2 Cells on the Microperforated Ti Membrane

We compared the morphology and proliferation of C3H10T1/2 cells on the head and tail sides of the Ti membrane using SEM (Figure 4). Many cells on the head side migrated to and accumulated on the microperforated holes on the 1st day. On the other hand, the cells on the tail side adhered to burr edges around the square pores on the surface with pseudopodia.

### 3.3. Distribution of C3H10T1/2 Cells on the Microperforated Ti Membrane

We examined the distribution of cells on the head and tail sides of the Ti membrane. The cells were seeded on the head side and cultured for 1 day. To compare with the culture dish, the same number of cells was seeded in an empty well. The cells on the head side adhered to the entire surface of the substrate, whereas the expression of F-actin was higher on the microperforated area than outside of it (Figure 5). The number of cells on the head side decreased slightly compared to the well. The cells on the tail side were distributed in the micropores. Furthermore, a negligible amount of F-actin was detected outside of the micropores.

### 3.4. Expansion of C3H10T1/2 Cells in the Multi-Layered Culture

We evaluated the capability of the Ti membrane to transfer cells to neighboring membranes. Each Ti membrane had its unevenness, and in the stacking assembly, a gap of approximately 0–500 μm formed between the Ti membranes. The cells migrated to both sides of the membrane (Figure 6A). The colonies formed in the surrounding area within the microperforated area. On the head side, the expression of F-actin was lower outside of the microperforated area than in the micropore area, which was in contrast to the distribution of nuclei; however, the microperforated area did not affect the expression of F-actin on the tail side (Appendix A). As shown in Figure 6B, the upper layers had more cells than the lower layers. A similar number of cells were observed in the layers +2 and ±0. In each layer, significant differences in cell numbers were noted between the head and tail sides, except for the layer ±0. The distribution of cells on Ti membranes after the second multi-layered culture is shown in Figure 7. The cell migration was observed on all substrates except the tail side of layer −2−2 (Figure 7A). The growth patterns in the second multi-layered culture had the same characteristics as in the first multi-layered culture. The substrates cultured with layer +2 ± 0 had slightly more cells than those cultured with layer −2±0 (Figure 7B).

### 3.5. Transfer of C3H10T1/2 Cells from the Ti Membrane to a Conventional Culture Plate

We focused on the transfer of cells from the Ti membrane to a conventional culture plate. The cells on the substrates migrated to and grew on a 12-well culture plate, except from layer +2−2 (Figure 8). The cell proliferation on the wells occurred at the same position as on the Ti membrane as they migrated from the neighboring substrate. The cell debris was detected in the wells cultured with layer +2−2.

## 4. Discussion

Pure titanium, a material of the Ti membrane, has a stable biocompatibility that can be applied to implantable medical devices of artificial joints and dental roots [19]. The property of this material surface was not an obstacle for cell attachment (Figure 5). The surface conditions of micro/nano-ordered roughness in biocompatible materials were not harmful to cell culture but instead promoted cell adhesion and proliferation [20].

The Ti membrane with a precise 25 μm square hole array at 75 μm intervals preserved the plane area for cell proliferation. The microtopographical architecture of scaffolds produces a biomimetic structure and influences cell activity [21], which is significant for tissue engineering applications [22]. Ti’s micron-order surface morphology promotes the cell’s adhesion and migration [23]. After punching a micron-sized hole array in the Ti membrane, numerous cutting edges were observed on the head side and apophyses of burrs around the edges on the tail side. The bodies of C3H10T1/2 cells cultured on the tail side surrounded the burrs. This microtopographical profile was an anchor to promote cell adhesion to the substrate.

The migration of cells on multi-layered Ti membranes is shown in Figure 9. These cells spread on the plane surface of the substrates regardless of their vertical orientation. The uniformed through-hole array provided 25 μm-wide pathways for cell movement. The distance of the pathways within the membrane enabled the cells to easily transfer to the surfaces of the opposite side and a proximal substrate. Repeating the above processes enabled long-term cell proliferation without enzymatic or chemical dissociation from the substrates. The time taken for the cells to transfer between the layers depends on the speed of cell migration and the distance between the contact points made by the layers. Approximately 0–5 cell colonies were observed on the membranes in Figure 6 and Figure 7. They are the areas where cells transfer between layers. The location of contact points between the layers is irregular, which may lead to instability in cell migration in Figure 8. Designing the Ti membrane to create more contact points between the layers may contribute to more efficient and stable cell migration.

In cell cultures, the physical properties of the culture substrate significantly affect cell attachment, which characterizes cell migration [24]. An increase in substrate stiffness efficiently drives cell adhesion and proliferation [25,26]. Ti is a commonly used biomaterial because of its superior biocompatibility [27] and high weight tolerance of physical loads, such as those placed on dental and orthopedic bones in the medical field [28,29]. Major commercial substrates on cell culture dishes include polymer plastics, including polystyrene, polyethylene terephthalate, and polypropylene. When these plastics are formed as a thin film of less than 30 μm and include a porous structure, the hardness of the substrate is significantly reduced. Therefore, the substrate is deformed due to the traction forces of the cells [30,31]. In the present study, the thin 10-μm Ti substrate was stable against the adhesion of numerous cells in the confluent stage. On the tail side of the membrane, the nano-topological tips of protrusions around the aperture acted as anchors for pseudopodia.

The Ti membrane becomes uneven during the fabrication process because of its rigidity. The ruggedness of the membrane created space for cell proliferation and contact points for the migration of cells to adjacent Ti membranes or the conventional culture vessel. However, these culture processes did not employ the continuous perfusion of a cell-conditioned medium in the column. The increase observed in the population of cells at the upper layer of the membrane may have been due to the gradient of nutrients and waste products from the cells in the medium [32]. Compared with a static culture, the perfusion method employing flow dynamics [33] is expected to promote the mass transport of nutrients and metabolism in scaffolds and overcome the diffusion limitations of a static culture [34,35,36].

Applying a microporous structure to the ultra-thin biocompatible metal produced a potential substrate enabling cell–cell contact, such as 3D cell proliferation, because of spatial diversity through assembly in a multi-layered structure. These conditions resembling 3D scaffolds may reproduce in vivo models more precisely than a 2D culture on a plane substrate [37]. Each membrane in the multi-layered culture must be utilized as a source for cultivation on the next stacked layer. Therefore, stacking Ti membranes enables the repeated migration of cells to the vacant membrane, which is reminiscent of reproducing clones by suckering in plants.

MSCs are highly proliferative due to their regenerative potential with pluripotent capacity. In clinical practice, MSCs are transferred into human biotissues in conjugation with scaffolds of specific biomaterials [38]. An ideal scaffold for MSCs in tissue engineering must act as a carrier to the recipient site and as a culture substrate to promote cell proliferation. The advantage of this method using a Ti microporous membrane as the culture substrate is that it provides compatibility between 2D and 3D environments. It was possible to return incubated cells from the Ti substrate to the conventional culture dish. The potential for cells to migrate from the Ti membrane to other substrates may be helpful in cell transplantation medicine.

Enzyme-free passaging, which has already been reported, allowed the cells to transfer from one substrate to another. However, these methods deviate from the conventional culture method of manipulation techniques because they do not have the plane substrate obtained with standard dish culture methods [39,40]. The Ti membrane can provide sufficient plane surface area for cell growth. Furthermore, this cultivation system may be operated in conventional culture vessels without additional tools. The conventional passaging process employing dissociation with trypsin provides a higher viability rate than enzyme-free dissociation buffer in MSC cultures [3]. However, the survival rate of MSCs on the Ti membrane was not evaluated in the present study. Establishing continuous proliferation without a dissociation process, similar to the environment in living biotissues, will facilitate the maintenance of interfaces between cells. Further research is needed to investigate whether the Ti membrane applies to other cell populations, such as primary cells.

## 5. Conclusions

A pure titanium membrane with unified 25 μm square holes at 75 μm intervals created by mechanical punching was applied as a scaffold for cultivating C3H10T1/2 cells. The pathway of pierced holes through the 10 μm-thick membranes allowed access to the cells between both sides. This membrane property allowed transferring the cells to the neighboring membranes stacked above and below. In the culture conditions using multi-layered membranes, a continuous proliferation of the cells was established without an enzymatic dissociation process. The Ti membrane has the potential as a scaffold for the C3H10T1/2 cells.

## Figures and Tables

**Figure 1 materials-16-01165-f001:**
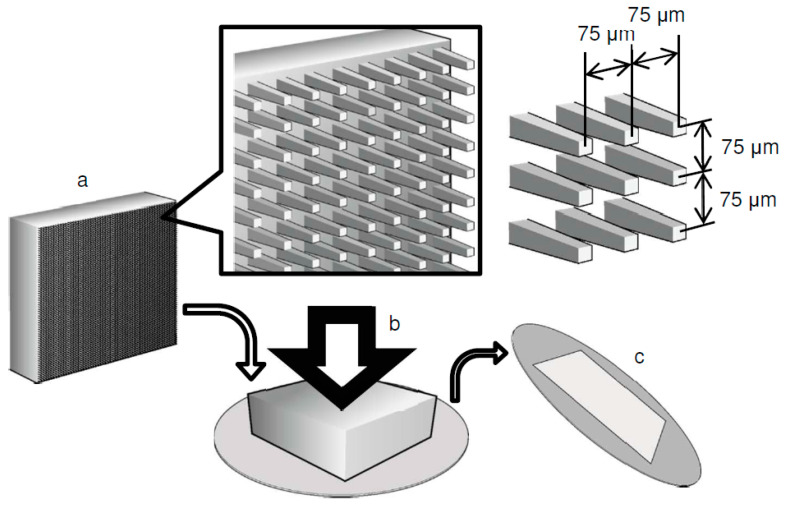
The fabrication process of the microperforated titanium (Ti) membrane: (**a**) Frog pin-type microarrayed pin needle punch (MPNP) was prepared with unified square needles evenly arrayed in 75 μm intervals. (**b**) The MPNP punched the Ti sheet with a servo press (1500 kg/cm^2^). (**c**) Ti membranes were formed with a precise array of 17,700 holes in a 10 × 10 mm area.

**Figure 2 materials-16-01165-f002:**
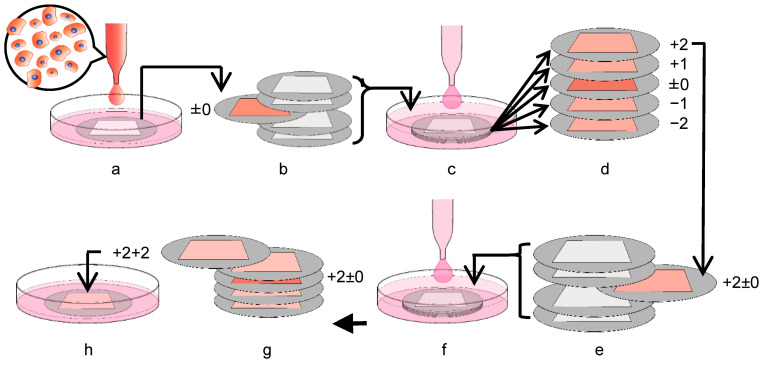
The scheme of multi-layered culture systems in the present study: (**a**) The C3H10T1/2 cells were seeded on a blank substrate and cultured for 24 h. (**b**) The substrate with C3H10T1/2 cells was transferred into the stacking assembly with 4 blank substrates without pressing. The membrane with seeded cells was defined as “±0.” (**c**) The multi-layered 3D substrates, including layer “±0,” were incubated for 10 days (first multi-layered culture). (**d**) In the first multi-layered culture, the upper layers were assigned positive numbers, whereas the lower layers were assigned negative numbers. (**e**) After the first multi-layered culture, each substrate moved into the other stacking assembly. This schematic shows that layer “+2” was used as a new cell source, defined as “+2±0”. (**f**) The new stacking assembly was incubated for 10 days (second multi-layered culture). (**g**) The layers above “+2±0” were assigned positive numbers (+2+1, +2+2), whereas the lower layers were assigned negative numbers (+2−1, +2−2). (**h**) After the second multi-layered culture, the substrates were isolated and cultured in a conventional culture vessel.

**Figure 3 materials-16-01165-f003:**
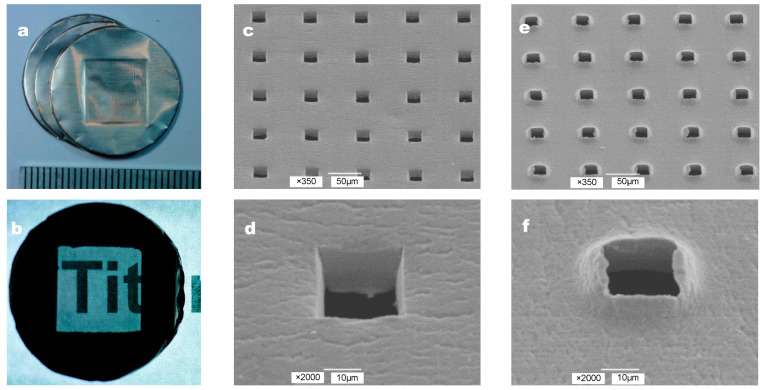
The form of the Ti membrane: (**a**) Approximately 17,700 microneedles in a 10 mm square block pierced the disc with a diameter of 22 mm to form hole arrays. (**b**) The partial transparency was preserved in the microperforated area, and an image was captured through the membrane. (**c**) The surface structures of microperforated arrays in Ti membranes using scanning electron microscopy. An overview of the 25 μm × 25 μm square aperture on the head side (the scale bar represents 50 μm). (**d**) A higher magnification view of an aperture on the head side (the scale bar represents 10 μm). (**e**,**f**) Two different magnification views of the tail side (the scale bars represent 10 or 50 μm).

**Figure 4 materials-16-01165-f004:**
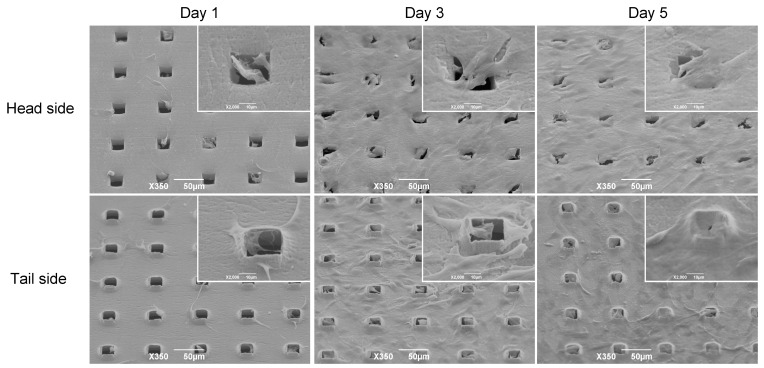
The visualization of cellular patterns on the microperforated substrate using scanning electron microscopy (SEM). The C3H10T1/2 cells were cultured on the substrate over a 5-day period. The cells cultured on the head side are shown in the top row, and those cultured on the tail side are shown in the bottom row. The SEM images are shown at 2 magnifications (the scale bars represent 10 or 50 μm).

**Figure 5 materials-16-01165-f005:**
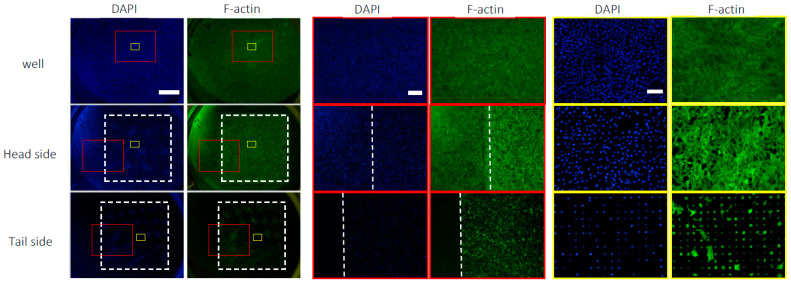
The fluorescence images of C3H10T1/2 cells cultured for 24 h on the head side and in the blank well. F-actin was detected by immunostaining (green) and nuclei were visualized by staining with DAPI (blue). The top row shows images on the head side, and the bottom row shows those on the tail side. The images are shown at 3 magnifications (the scale bars represent 150, 750, or 3000 μm). The microperforated areas are inside the white box in low magnification images, the right side of the white line in middle magnification images, and in the entire area in high magnification images.

**Figure 6 materials-16-01165-f006:**
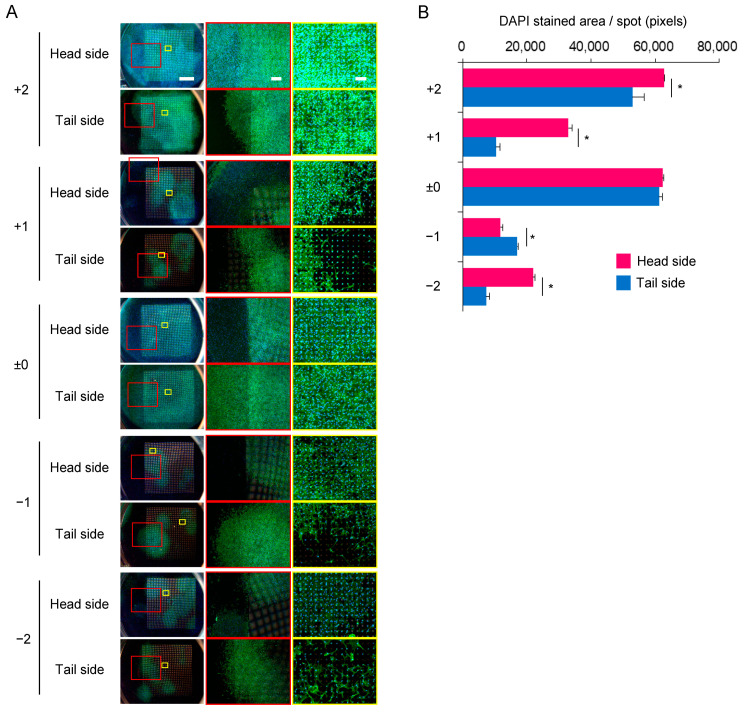
The migration of C3H10T1/2 cells to neighboring Ti membranes. The substrate on which the cells were incubated for 24 h was placed between four new membranes and cultured for 10 days (first multi-layered culture). The membrane with seeded cells was defined as “±0”. The upper layers were assigned positive numbers, whereas the lower layers were assigned negative numbers. (**A**) The cells were stained for F-actin (green) and nuclei (blue). The images are shown at three magnifications (the scale bars represent 150, 750, or 3000 μm). (**B**) The stained areas of nuclei in panel A were quantified. The significance is represented as * *p* < 0.05.

**Figure 7 materials-16-01165-f007:**
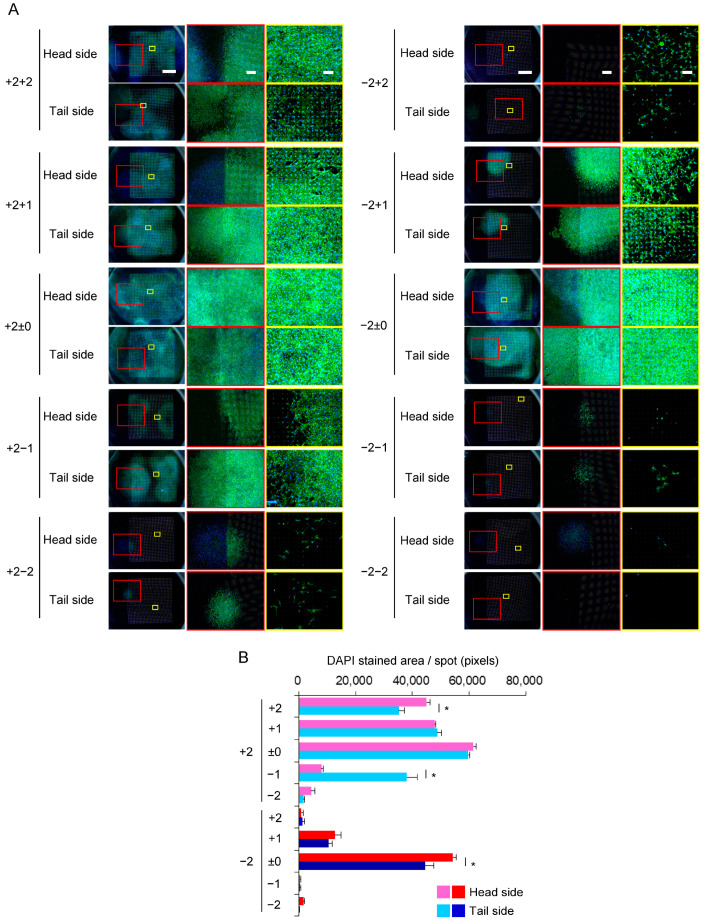
The expansion of C3H10T1/2 cells in multi-layered cultures: After the first multi-layered culture, the highest and lowest layers of membranes were separately placed between four new membranes and incubated for another 10 days (second multi-layered culture). The highest layer in the first multi-layered culture was defined as “+2±0,” and the lowest layer was “−2±0.” The upper layers were assigned positive numbers, whereas the lower layers were assigned negative numbers in the second multi-layered culture. (**A**) The cells were stained for F-actin (green) and nuclei (blue). The images are shown at 3 magnifications (the scale bars represent 150, 750, or 3000 μm). (**B**) The stained areas of nuclei in panel A were quantified. The significance is represented as * *p* < 0.05.

**Figure 8 materials-16-01165-f008:**
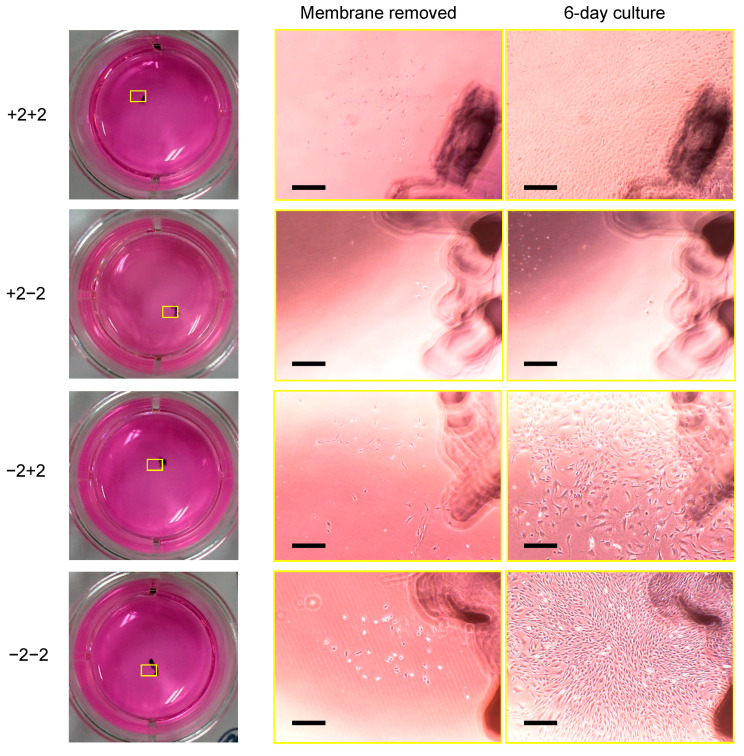
The outgrowth of C3H10T1/2 cells from the Ti membrane to a culture dish: following the second multi-layered culture, the top and bottom substrates were separately cultured in a new culture plate for 3 days. After removing the substrates, the cells migrating to the plate were cultured for 6 days. The selected areas (yellow boxes) in the new culture plate were compared between when the substrates were removed and when cells were cultured for 6 days (the scale bar represents 500 μm).

**Figure 9 materials-16-01165-f009:**
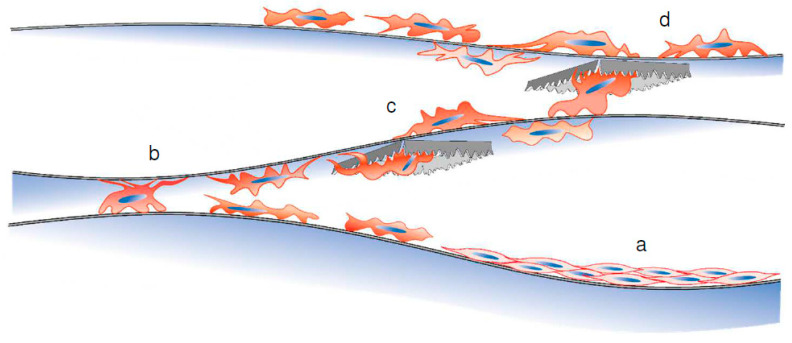
The schema of cell proliferation in the multi-layered Ti membrane. The C3H10T1/2 cells seeded on the original membrane (layer “±0”) grew in the space between layers “±0” and “+1” (a). The cells climbed to layer “+1” at the contact point between the layers “±0” and “+1” (b). The cells divided and expanded around layer “+1” and migrated to the opposite side of the membrane via nano-sized pathways (c). The cells rose to layer “+2” at the contact point between the layers “+1” and “+2” (d). The burrs around the pores did not interrupt cell migration to the upper layer.

## Data Availability

Not applicable.

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
