# Peer review of "Serial Cultivation of an MSC-Like Cell Line with Enzyme-Free Passaging Using a Microporous Titanium Scaffold"

_materials, 2023, doi:10.3390/ma16031165_

Round 1

Reviewer 1 Report

Overall the article is good and appropriate for the readers on the subject. The methodology outlined is appropriate.

My biggest comment here is that, the conclusion is very short and shallow despite of the good results presented.

Author Response

Overall the article is good and appropriate for the readers on the subject. The methodology outlined is appropriate.
My biggest comment here is that, the conclusion is very short and shallow despite of the good results presented.
Answer: Thank you very much for giving your positive interest in this research. According to your suggestion, we have added a sentence explaining the core concept of the study. We would like to develop methods to bring cell proliferation closer to actual tissue growth.

Reviewer 2 Report

This is an interesting study with a goal to develop new type of biomaterial for enzyme free cell culture passaging. This work can be of interest to a broad readership. However, the way the work was presented do not support the fact that the method of the study ultimately achieves the goal. I was very confused about what is really attempted here. Is the goal to have a different system than a 2D surface culture or is it a 3D system that will be utilized for using in combination with 2D system? Either way, please explain the goal and how much is actually achieved to reach that goal.

I am also not convinced about the need for doing this. The harmful effect of enzyme mediated passaging is only mentioned with one citation. Some rigorous previous results must be presented to make a case for this work.

I am also not convinced that the new technique can really does not have other harmful effect on the cells. I would like to see comparative results of the harmful effect of the enzyme vs the new platform. At the minimum, a viability study is required.

Finally, the cell line used is not a standard MSC cell type. Therefore, it is not clear how useful the technique can be used for a broad application with other cell types.

Author Response

This is an interesting study with a goal to develop new type of biomaterial for enzyme free cell culture passaging. This work can be of interest to a broad readership. However, the way the work was presented do not support the fact that the method of the study ultimately achieves the goal. I was very confused about what is really attempted here. Is the goal to have a different system than a 2D surface culture or is it a 3D system that will be utilized for using in combination with 2D system? Either way, please explain the goal and how much is actually achieved to reach that goal.
Answer: Thank you very much for your constructive suggestion. Traditionally, 2D culture and 3D culture have been performed using different materials and methods. In this method, there is little difference in materials and techniques between 2D and 3D cultures. The only difference between 2D and 3D conditions is that the culture substrates are single or multiple stacks. The membrane scaffold developed in this project has the potential to culture cells in the same way as 2D planar culture. Also, by stacking and layering this scaffold, it seems possible to function as a 3D scaffold. You would choose these 2D and 3D properties according to your concept. A serial culture method was already developed that avoids enzyme passaging by using beads and fibers as cell culture substrates. However, these methods deviate significantly from common plate culture methods. The essence of this experiment was to realize a serial cultivation without enzymatic passaging by incorporating spatial redundancy on the conservative culture method. Therefore, we originally produced the Ti membrane to establish a new method of the serial cultivation.
We have slightly correct sentences according to the purpose of this research that a new method could perform continuously growing cells without passaging by enzymes, acids, or scratch detachment. In order to achieve this non-enzymatic cultivation method, scaffolds are used in two specifications: 2D planar culture and 3D stacking culture.
For clinical applications in the future, we plan to stylize this scaffold as 3D culture and apply it as a carrier to deliver cells in vivo.

I am also not convinced about the need for doing this. The harmful effect of enzyme mediated passaging is only mentioned with one citation. Some rigorous previous results must be presented to make a case for this work.
Answer: According to your suggestion we have added the sentence and references.
I am also not convinced that the new technique can really does not have other harmful effect on the cells. I would like to see comparative results of the harmful effect of the enzyme vs the new platform. At the minimum, a viability study is required.
Answer: According to your suggestion we indicated the facility and reliability of the new culture by adding the figure of the conventional culture dish as a positive control in figure 5.
Finally, the cell line used is not a standard MSC cell type. Therefore, it is not clear how useful the technique can be used for a broad application with other cell types.
Answer: As you mentioned, it is the study limitation of our study. We have plans for the cultivation of mouse bone marrow stem cells.

Author Response

The paper shows an interesting concept that can be useful for many cell lines and especially for metabolic activity or differentiation of stem cells. It could be beneficial maybe as a future study to compare this method on more cellular behaviors and differentiation aspects. In addition, the effect of micro-holes shape and distances could also be studied. Overall the manuscript is well written and I suggest to be published in the Materials journal. Although there are a few questions I would like to ask before accepting the paper.

Answer: Thank you very much for your constructive suggestion. We corrected the sentences below.

Page 1 section 1:
The introduction part is too short and does not cover the other enzyme-free methods that has been used in the literature.
Answer: According to your suggestion we have added the sentence and references.

Page 3 line 100:
How adherent is the Ti surface? When mentioned “Ti membranes were removed and cells on the culture plate were incubated …” Have you applied any forces to remove the cells? Or after removal how many cells were left on the Ti membrane? In general, how strong is the adhesion of the cells to the substrate?
Answer: The cells on the 12-well culture plate with the Ti membrane removed were ones that transferred from the membrane to the culture plate for 3-day culture. Even if we removed the Ti membrane, the adherent cells on the membrane were left on the membrane. We have exchanged the sentence to clarify the meaning.

Page 6 section 3.3 line 176:
It is stated that distribution of cells seeded on the head side of the Ti membrane was examined, but the images show both sides of membrane.
Answer: We observed the cells on both sides of the Ti membrane which we seeded only on the head side of the membrane. We rewrote the sentence according to your suggestion.

Page 6 section 3.4 line 190:
How did you measure the gap distance of Ti membranes?
Answer: 5 layers of the 10 μm-thick membranes were about 2 mm-high. Therefore, we regarded the gap between each membrane as 0-500 μm.

Page 6 section 3.4 line 192:
Its mentioned that “On the head side, F-actin expression was lower outside of the microperforated area than in the micropore area, which was in contrast to the distribution of nuclei”. How could you comment on the distribution of nuclei around micropores? Again, at line 194 its stated that “the microperforated area did not affect the expression of F-actin on the tail side.” Please clarify your reasoning about this statement.
Answer: According to your suggestion we have added Supplementary Materials.

Page 9 section 3.5 line 223:
The layer +2-2 which showed no cells, was this behavior observed in all replicates of layer +2-2?
Answer: Several dishes were usually difficult to find the cell growth when the Ti membrane was removed after incubation for 3 days. It is confirmed that all layers let cells transfer to dishes with longer incubation periods. According to your suggestion, we have added the sentence to the discussion.

Round 2

Reviewer 2 Report

The manuscript is significantly improved. I am happy with the present form.